# Bayesian Quadrature for Neural Ensemble Search

**Saad Hamid**                                                    *saad@robots.ox.ac.uk*
*University of Oxford*

**Xingchen Wan**                                                  *xwan@robots.ox.ac.uk*
*University of Oxford*

**Martin Jørgensen**                                             *martinj@robots.ox.ac.uk*
*University of Oxford*

**Binxin Ru**                                                    *robin@robots.ox.ac.uk*
*University of Oxford*

**Michael Osborne**                                              *mosb@robots.ox.ac.uk*
*University of Oxford*

**Reviewed on OpenReview:** *https://openreview.net/forum?id=T5sXdAO3EQ*

## Abstract

Ensembling can improve the performance of Neural Networks, but existing approaches struggle when the architecture likelihood surface has dispersed, narrow peaks. Furthermore, existing methods construct equally weighted ensembles, and this is likely to be vulnerable to the failure modes of the weaker architectures. By viewing ensembling as approximately marginalising over architectures we construct ensembles using the tools of Bayesian Quadrature – tools which are well suited to the exploration of likelihood surfaces with dispersed, narrow peaks. Additionally, the resulting ensembles consist of architectures weighted commensurate with their performance. We show empirically – in terms of test likelihood, accuracy, and expected calibration error – that our method outperforms state-of-the-art baselines, and verify via ablation studies that its components do so independently.

## 1 Introduction

Neural Networks (NNs) are extremely effective function approximators. Their architectures are, however, typically designed by hand, a painstaking process. Therefore, there has been significant interest in the automatic selection of NN architectures. In addition to a search strategy, this involves defining a search space from which to select the architecture – a non-trivial which is also an active area of research. Recent work shows ensembles of networks of different architectures from a given search space can outperform the *single* best architecture or ensembles of networks of the same architecture (Zaidi et al., 2022; Shu et al., 2022). Finding the single best architecture is typically referred to as Neural Architecture Search (NAS) (Zoph & Le, 2016; Elsken et al., 2019; He et al., 2021). Such ensembles improve performance on a range of metrics, including the test set's predictive accuracy, likelihood, and expected calibration error. The latter two metrics measure the quality of the model's uncertainty estimates, which can be poor for single architectures in some cases (Guo et al., 2017). Performant models in these metrics are crucial for systems which make critical decisions, such as self-driving vehicles. Ensemble selection is an even more difficult problem to tackle manually than selecting a single architecture, as it requires a combinatorial search over the same space. Hence, interest in methods for automatic ensemble construction is growing. This paper targets exactly this problem.

Conceptually, Neural Ensemble Search (NES) algorithms can be split into two stages. The first is the *candidate selection* stage, which seeks to characterise the posterior distribution, $p(\alpha \mid D)$, given the training

data $D$, over architectures from a given search space $\alpha \in \mathcal{A}$. Multiple approaches have been proposed. One such is an evolutionary strategy which seeks the modes of this distribution (Zaidi et al., 2022). An alternative is training a "supernet" and using it to learn the parameters of a variational approximation to this distribution (Shu et al., 2022). This involves evaluating the likelihood of a set of architectures from the search space, an evaluation which requires first training the architecture weights. The second stage is *ensemble selection*, where the ensemble members are selected from the candidate set and each member's weight is chosen. Several approaches have also been suggested for ensemble selection, such as beam search and sampling from the (approximate) posterior over architectures.

In this work, we investigate novel approaches to both stages of a NES algorithm. We view ensembling, the averaging over architectures, as marginalisation with respect to a particular distribution over architectures. When this distribution is the posterior over architectures, we are taking the hierarchical Bayesian approach. The key advantage of this approach is the principled accounting of uncertainty, which also improves accuracy by preventing overconfidence in a single architecture. Additionally, this paradigm allows us to bring the tools of Bayesian Quadrature to bear upon the problem of Neural Ensemble Search. Specifically, the contributions of this work are as follows:[1]

- We propose using an acquisition function for adaptive Bayesian Quadrature to select the candidate set of architectures to train. It is from this candidate set that the ensemble members are later selected.

- We show how recombination of the approximate posterior over architectures can be used to construct a weighted ensemble from the candidate set.

- We undertake an empirical comparison of our proposals against state-of-the-art baselines. Additionally, we conduct ablation studies to understand the effect of our proposals for each stage of the NES pipeline.

## 2 Background

### 2.1 Neural Architecture Search (NAS)

NAS aims to automatically discover high-performing NN architectures and has shown promising performance in various tasks (Real et al., 2017; Zoph et al., 2018; Liu et al., 2019a). It is typically formulated as an optimisation problem, i.e. maximising some measure of performance $f$ over a space of NN architectures $\mathcal{A}$,

$$\alpha_* = \text{argmax}_{\alpha \in \mathcal{A}} f(\alpha). \tag{1}$$

Elsken et al. (2019) identify three conceptual elements of a NAS pipeline: a search space, a search strategy, and a performance estimation strategy.

The first part of a NAS pipeline – the search space – is the way in which the possible space of NN architectures is defined. In this paper, we require the search space to be such that a Gaussian Process (GP) can be defined upon it. In particular, we focus on the two types of search space most common in the literature. The first is a *cell-based* search space, which consists of architectures made by swapping out "cells" in a fixed macro-skeleton (Pham et al., 2018; Dong et al., 2021; Liu et al., 2019b). These cells are represented as directed acyclic graphs where each edge corresponds to an operation from a pre-defined set of operations. Typically, the macro-skeleton will be structured so that repeating copies of the same cell are stacked within the macro-skeleton. This structure allows for the representation of the architecture by the corresponding cell. The second is a *macro* search space defined by varying structural parameters such as kernel size, number of layers, and layer widths. An example of such a search space is the *Slimmable network* (Yu et al., 2019; Yu & Huang, 2019) on the MobileNet search space (Sandler et al., 2018): the largest possible network is trained and all other networks in the search space are given as sub-networks or "slices".

A NAS pipeline's second phase is the search strategy. This is a procedure for selecting which architectures to query the performance of. All strategies will exhibit an exploration-exploitation trade-off, where exploration

---

[1]An implementation of our proposals can be found at https://github.com/saadhamidml/bq-nes.

is covering the search space well, and exploitation is selecting architectures that are similar to the well-performing architectures in the already queried history.

The final element of a NAS pipeline is the performance estimation strategy, which is the method for querying the performance of a given architecture. Typically, this is done by training the NN weights, given the architecture, on a training dataset, and evaluating its performance on a validation set. This demands significant computation and practically limits the total number of architecture evaluations available to the search strategy. However, for some search spaces – where network weights are shared – performance estimation is considerably cheaper.

There exists a large body of literature devoted to NAS, pursuing a range of strategies such as one-shot NAS (Liu et al., 2019b; Xu et al., 2020; Chen et al., 2019; Yu et al., 2020; Bender et al., 2018), evolutionary strategies Real et al. (2017); Liang et al. (2018); Liu et al. (2021), and Bayesian Optimisation.

## 2.2 Bayesian Optimisation for Neural Architecture Search

An effective approach to NAS is Bayesian Optimisation (BO) (Kandasamy et al., 2019; White et al., 2020; Ru et al., 2021; Wan et al., 2022; Shi et al., 2020; Zhou et al., 2023). On a high level, BO models the objective function $f$ and sequentially selects where to query next based on an acquisition function, with the goal of finding the optimal value of the objective function in a sample-efficient manner. Typically, the objective function is modelled using a Gaussian Process (GP) – a stochastic process for which all finite subsets of random variables are joint normally distributed (Rasmussen & Williams, 2006).

A GP is defined using a mean function $m(\alpha)$ that specifies the prior mean at $\alpha$, and a kernel function $k(\alpha, \alpha')$ that specifies the prior covariance between $f(\alpha)$ and $f(\alpha')$. The posterior, conditioned on a set of observations $A = \{(\alpha_i,)\}_i^N$ and $y = [f(\alpha_1), \dots, f(\alpha_N)]^T$, is also a GP with moments

$$m_A(\cdot) = m(\cdot) + K_{\cdot A} K_{AA}^{-1} \left( y - m(\alpha) \right) \qquad \text{and} \tag{2}$$

$$k_A(\cdot, \cdot') = K_{\cdot \cdot'} - K_{\cdot A} K_{AA}^{-1} K_{A \cdot}, \tag{3}$$

where $K_{XY}$ indicates a matrix generated by evaluating the kernel function between all pairs of points in the sets $X$ and $Y$. The prior mean function $m$ is typically set to zero.

Ru et al. (2021) showed that the Weisfeiler-Lehman graph kernel (WL kernel) (Shervashidze, 2011) is an appropriate choice for modelling NN performance metrics on a cell-based NAS search space with a GP. To apply the WL kernel, a cell first needs to be represented as a labelled DAG. Next, a feature vector is built up by aggregating labels for progressively wider neighbourhoods of each node, and building a histogram of the resulting aggregated labels. The kernel is then computed as the dot product of the feature vectors for a pair of graphs.

A common acquisition function for BO is Expected Improvement (Garnett, 2021),

$$a_{EI}(\alpha) = \mathbb{E}_{p(f|D)} \Big[ \max \big( f(\alpha) - f(\hat{\alpha}), 0 \big) \Big] \tag{4}$$

where $\hat{\alpha}$ is the best architecture found so far. Using this acquisition function in conjunction with a GP using the WL kernel was shown by Ru et al. (2021) to be effective for NAS.

## 2.3 Neural Ensemble Search

Neural Ensemble Search (Zaidi et al., 2022) is a method for automatically constructing ensembles of a given size, $M$, from a NAS search space $\mathcal{A}$. First, a candidate set of architectures, $A \subset \mathcal{A}$, is selected using a regularised evolutionary strategy (NES-RE), or random sampling from the search space. The authors propose several ensemble selection methods to subsequently select a subset of $M$ architectures $A_M \subset A$. Of particular interest in this work are Beam Search (BS) and Weighted Stacking (WS).

BS initially adds the best-performing architecture to the ensemble and greedily adds the architecture from the candidate set (without replacement) that most improves the validation loss of the ensemble. WS optimises

the ensemble weights over the whole candidate set on the validation loss (subject to the weights being non-negative and summing to one). The members with the highest $M$ weights are included in the ensemble, and their corresponding weights renormalised. The authors compare BS to WS on the CIFAR-10 dataset, and find performance in terms of the log-likelihood of the test set to be better for BS for small ensembles, but similar for larger ensembles.

Neural Ensemble Search via Bayesian Sampling (Shu et al., 2022) approximates the posterior distribution over architectures $p(\alpha \mid D)$ with a variational distribution of the form $q(\alpha) = \prod_i q_i(o \mid \boldsymbol{\theta}_i)$, where $i$ iterates over the connections within a cell, $o$ is the operation for connection $i$, and $\boldsymbol{\theta}_i$ are the variational parameters for $q_i$. The form of $q_i$ is chosen to be a softmax over $\boldsymbol{\theta}_i$. The ensemble is then selected by using Stein Variational Gradient Descent with Regularised Diversity to select a diverse set of $M$ samples from (a continuous relaxation of) the variational distribution.

DeepEnsembles (Lakshminarayanan et al., 2017) seek to approximately marginalise over the parameters of a given NN architecture. The architecture is trained from several random initialisations, and the ensemble makes a prediction as an equally weighted sum of these. Hyper-deep ensembles (Wenzel et al., 2020) build on this idea by training architectures from several randomly selected hyperparameter settings. They then construct an ensemble by using Beam Search, with replacement.

Relatedly, there has been interest in ensembling for improving uncertainty calibration in the context of Reinforcement Learning (Osband et al., 2018; Dwaracherla et al., 2022). Additionally, methods for marginalising over the parameters of a fixed architecture such as Shui et al. (2018); He et al. (2020); D'Angelo & Fortuin (2021) frequently require constructing ensembles. Such methods are orthogonal to our work, which seeks to construct ensembles of different architectures, rather than ensembles of different parameter settings of the same architecture.

### 2.4 Bayesian Quadrature

Bayesian Quadrature (BQ) (O'Hagan, 1991; Minka, 2000) is a probabilistic numerical (Hennig & Osborne, 2022) integration technique that targets the computation of $Z = \int f(\cdot)d\pi(\cdot)$ based on evaluations of the integrand $f$ (assuming a given prior $\pi$). Similar to BO, it maintains a surrogate model over the integrand $f$, which induces a posterior over the integral value $Z$. BQ also makes use of an acquisition function to iteratively select where next to query the integrand.

The surrogate model for BQ is usually chosen to be a GP, and this induces a Gaussian posterior over $Z \sim \mathcal{N}(\mu_Z, \sigma_Z)$. The moments of this posterior are given by

$$\mu_Z = \int K(\cdot, X)d\pi(\cdot)K_{XX}^{-1}f, \quad \text{and} \tag{5}$$

$$\sigma_Z = \int K(\cdot, \cdot') - K(\cdot, X)K_{XX}^{-1}K(X, \cdot)d\pi(\cdot)d\pi(\cdot'), \tag{6}$$

where $X$ is the set of query points, and $f$ are the corresponding integrand observations. Note that the posterior mean $\mu_Z$ takes the form of a quadrature rule – a weighted sum of function evaluations $\sum_i w_i f(x_i)$ where $w_i$ are the elements of the vector $\int K(\cdot, X)d\pi(\cdot)K_{XX}^{-1}$.

Frequently, the integrand of interest is non-negative. Important examples of such integrands are likelihood functions (which are integrated with respect to a prior to compute a model evidence) and predictive densities (which are integrated with respect to a posterior to compute a posterior predictive density). Warped Bayesian Quadrature (Osborne et al., 2012; Gunter et al., 2014; Chai & Garnett, 2019) allows practitioners to incorporate the prior knowledge that the integrand is non-negative into the surrogate model. Of particular interest in this work will be the WSABI-L model (Gunter et al., 2014), which models the square root of the integrand with a GP, $\sqrt{2(f(x) - \beta)} \sim \mathcal{GP}(\mu_D(x), \Sigma_D(x, x'))$. This induces a (non-central) chi-squared distribution over $f$, which can be approximated with a GP, with moments

$$m(x) = \beta + \frac{1}{2}\mu_D(x)^2, \tag{7}$$

$$k(x, x') = \mu_D(x)\Sigma_D(x, x')\mu_D(x'). \tag{8}$$

Gunter et al. (2014) established, empirically, that the uncertainty sampling acquisition function works well for Bayesian Quadrature. This acquisition function targets the variance of the integrand

$$a_{US}(x) = \Sigma_D(x, x)\mu_D(x)^2\pi(x)^2. \tag{9}$$

This naturally trades off between exploration (regions where $\Sigma_D(x, x)$ is high), and exploitation (regions where $\mu_D(x)$ is high – most of the volume under the integrand is concentrated here).

Just as BO is a natural choice for NAS – an expensive black-box optimisation problem – so BQ is a natural choice for NES – an expensive black-box marginalisation problem. It is this realisation that inspires our proposals in Section 3.

## 2.5 Recombination

Typically, Neural Ensemble Search requires selecting a subset of the trained architectures to build the ensemble, as making predictions with the whole candidate set is too computationally burdensome. To achieve this within the Bayesian Quadrature framework will require reducing the support of the quadrature rule. This problem is referred to as recombination (Litterer & Lyons, 2012).

Given a non-negative measure supported on $N$ points $\{(w_n, x_n)\}_{n=1}^N$ where $w_n \geq 0$ and $\sum_{n=1}^N w_n = 1$, and $M - 1$ "test" functions $\{\phi_t(\cdot)\}_{t=1}^{M-1}$, it is possible to find a subset of $M < N$ points $\{x_n\}_{m=1}^M \subset \{x_n\}_{n=i}^N$ for which

$$\sum_{m=1}^M w_m \phi_t(x_m) = \sum_{n=1}^N w_n \phi_t(x_n) \tag{10}$$

for all $\phi_t$, with $w_m \geq 0$ and $\sum_{m=1}^M w_m = 1$ (Tchernychova, 2015).

For Kernel Quadrature, one can use the Nyström approximation of the kernel matrix to obtain a set of test functions (Hayakawa et al., 2022; Adachi et al., 2022). Using a subset, $S$, of $M - 1$ data points, the kernel can be approximated $\tilde{k}(x, x') = k(x, S)k(S, S)^{-1}k(S, x')$. By taking an eigendecomposition, $k(S, S) = U\Lambda U^T$, the approximate kernel can be expressed as

$$\tilde{k}(x, x') = \sum_t^{M-1} \frac{1}{\lambda_t}\left(u_t^T k(S, x)\right)\left(u_t^T k(S, x')\right) \tag{11}$$

where $u_i$ are the columns of $U$, and $\lambda_i$ the diagonal elements of $\Lambda$. We can then use $\phi_t(\cdot) = u_t^T k(S, \cdot)$ as test functions.

# 3 Bayesian Quadrature for Neural Ensemble Search

We decompose NES into two sub-problems:

1. The selection of a candidate set of architectures $\{\alpha_i\}_{i=1}^N = A \subset \mathcal{A}$ for which to train the architecture parameters.

2. The selection of a set of $M$ members from the candidate set to include in the ensemble, and their weights, $\boldsymbol{w} \in \mathbb{R}^M$.

We take novel approaches to each of these sub-problems, described respectively in the following two subsections. Algorithms 1, 2 and 3 summarise our propositions.

## 3.1 Building the Candidate Set

An ensemble's prediction is a weighted sum of the predictions of its constituent members, $A_M$, and this can always be viewed as approximating an expectation with respect to a distribution, $\pi$, over architectures,

$$\mathbb{E}_{\pi(\alpha)}\big[p(c \mid x, \alpha)\big] = \sum_{\alpha \in \mathcal{A}} p(c \mid x, \alpha)\pi(\alpha) \approx \sum_{\alpha \in A_M} p(c \mid x, \alpha)\pi_M(\alpha). \tag{12}$$

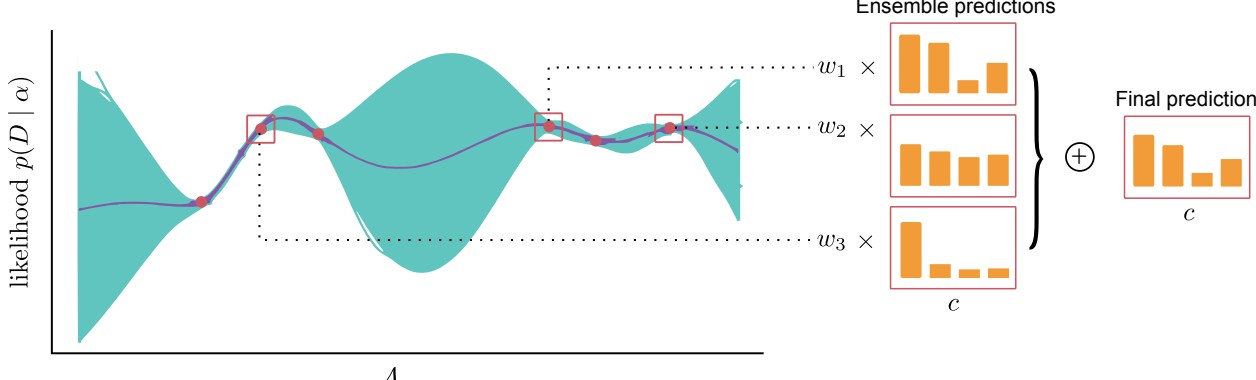

Figure 1: A schematic representation of our proposal. The plot on the left shows a Gaussian Process modelling the likelihood over the space of architectures. The architectures to train and evaluate the likelihood for are selected by maximising a Bayesian Quadrature acquisition function, as described in Section 3.1. One of the algorithms described in Section 3.2 is then used to select the subset of architectures to include in the ensemble, along with their weights. The final prediction is then a linear combination of the predictions of each ensemble member, as shown by the bar plots on the right (where each bar indicates the probability assigned to a particular class).

$p(c \mid x, \alpha, D)$ is the predictive probability assigned to class $c \in \{1, \dots, C\}$ by the architecture $\alpha$ (conditioned on the training data $D$) given the input $x \in \mathcal{X}$. The expectation in Equation (12) corresponds to a marginalisation over architectures. The set of architectures $A_M$ and their weights $\pi_M$ can be seen as a quadrature rule to approximate this marginalisation. When $\pi$ is the posterior over architectures $p(\alpha \mid D)$, we are performing hierarchical Bayesian inference, and the result is the posterior predictive distribution,

$$p(c \mid x, D) = \sum_{\alpha \in \mathcal{A}} p(c \mid x, \alpha, D) \, p(\alpha \mid D)$$
$$= \frac{\sum_{\alpha \in \mathcal{A}} p(c \mid x, \alpha, D) \, p(D \mid \alpha) \, p(\alpha)}{\sum_{\alpha \in \mathcal{A}} p(D \mid \alpha) \, p(\alpha)}. \tag{13}$$

By taking the view of ensembling as marginalisation, the practitioner has the ability to make their belief over $\mathcal{A}$ explicit. As the training data is finite, it is rarely appropriate to concentrate all of the probability mass of $\pi$ on a single architecture. Arguably, $\pi(\alpha) = p(\alpha \mid D)$ is the most appropriate choice of $\pi$ as it is the distribution implied by the prior and the architectures' ability to explain the observed data. The hierarchical Bayesian framework should offer the most principled accounting of uncertainty in the choice of architecture (a more concentrated $\pi$ should over-fit, a less concentrated $\pi$ should under-fit).

From Equation (13) we see that, to compute the posterior predictive, we need to compute $C$ sums of products of functions of the architecture and the architecture likelihoods. Intuitively, we expect a quadrature scheme that approximates well the sum in the denominator of (13) will also approximate the sum in the numerator well. Therefore, we propose using a Bayesian Quadrature acquisition function to build up the candidate set, as these architectures will form the nodes of a query-efficient quadrature scheme for (13) and so a good basis for an ensemble.

The likelihood of an architecture $p(D \mid \alpha)$ is not typically available, as this would require marginalisation over the NN weights, $w$, of the architecture. We instead approximate using the MLE, which is equivalent to assuming the prior of the architecture weights is a Dirac delta distribution at the maximiser of the (architecture weights') likelihood function.

$$p(D \mid \alpha) = \int p(D \mid w, \alpha) \, p(w \mid \alpha) dw \approx p(D \mid \hat{w}, \alpha), \tag{14}$$

$$\hat{w} = \mathrm{argmax}_w p(D \mid w, \alpha) p(w \mid \alpha). \tag{15}$$

Computing $p(c \mid x, \alpha, D)$ requires an analogous intractable marginalisation. We approximate it similarly, noting that it depends only indirectly on the training data, through the optimisation procedure, i.e.

$$p(c \mid x, \alpha, D) = \int p(c \mid x, w, \alpha, D) \, p(w \mid \alpha, D) dw$$

$$\approx p(c \mid x, \hat{w}, \alpha). \tag{16}$$

Concretely, we place a functional prior on the architecture likelihood surface, warped using the square-root transform, $\sqrt{2\big(p(D \mid \hat{w}, \alpha) - \beta\big)} \sim \mathcal{GP}$, and use uncertainty sampling to make observations of the likelihood at a set of architectures $\{\alpha_i\}_{i=1}^{N} = A \subset \mathcal{A}$.

This provides us with an estimate of the model evidence $Z = \sum_{\alpha \in \mathcal{A}} p(D \mid \hat{w}, \alpha) \, p(\alpha)$, which we denote $\hat{Z}$. The computation of this estimate requires Monte Carlo sampling to approximate sums of (products of) the WL-kernel over $\mathcal{A}$. Note this is far more feasible than approximating the original sums in Equation (13) with Monte Carlo sampling as $K(\alpha_j, A)$ is far cheaper to evaluate than $p(D \mid \hat{w}, \alpha_j)$ or $p(c \mid x, \hat{w}, \alpha_j)$ – either would require training architecture $\alpha_j$.

---

**Algorithm 1** Candidate set selection algorithm using a BQ acquisition function. Returns architectures $A = \{\alpha_i\}_{i=1}^{N}$ and their corresponding likelihoods $L = \{p(D \mid \hat{w}, \alpha_i)\}_{i=1}^{N}$.

$A, L \leftarrow \text{sample}(n, \mathcal{A})$           ▷ Initial samples.
$\theta \leftarrow \text{argmax}_\theta p(L \mid A, \theta)$           ▷ Optimise WL kernel.
**while** $i > 0$ **do**
     $\alpha \leftarrow \text{argmax}_{\alpha \in \mathcal{A}} \text{acquisition\_function}(\alpha, A, L, \theta)$
     $A \leftarrow \{A, \alpha\}$
     $L \leftarrow \{L, p(D \mid \hat{w}, \alpha)\}$
     $\theta \leftarrow \text{argmax}_\theta p(L \mid A, \theta)$
**end while**
**return** $A, L$

---

### 3.2 Selecting the Ensemble

In principle, the ensemble can be constructed using the weights provided by the quadrature scheme, as these weights naturally trade-off between member diversity and member performance. However, we wish to select a subset of the candidate set for the ensemble (as it is assumed that an ensemble of the whole candidate set is too costly to be practical for deployment). Concretely, we seek a subset $A_M \subset A$, along with weights $\boldsymbol{w} \in \mathbb{R}^M$ such that

$$p(c \mid x, D) \approx \sum_{n}^{N} \frac{1}{\hat{Z}} p(D \mid \alpha_n) \, p(\alpha_n) \, p(c \mid x, \alpha_n, D) + \epsilon \tag{17}$$

$$\approx \sum_{m}^{M} w_m p(c \mid x, \alpha_m, D) + \epsilon. \tag{18}$$

We expect $\epsilon$ to be small if regions of high likelihood have been well-explored by the acquisition function in the building of the candidate set. To select the weights $\boldsymbol{w}$ and the set $A_M$ we can use any recombination algorithm, using the Nyström approximation to generate the test functions, as described in Section 2.5, and the estimated posterior over architectures as the measure to recombine. We refer to this algorithm as Posterior Recombination (PR).

A second approach, which we refer to as Re-weighted Stacking (RS), is a modification of Weighted Stacking. Similar to WS, we optimise the weights of an ensemble of the whole candidate set to minimize the validation loss. The ensemble members are then chosen by selecting the members with the $M$ highest weights. However, rather than renormalising the corresponding weights, as suggested in Zaidi et al. (2022), we reallocate the weight assigned to excluded architectures proportionally to the relative covariance between them and the

ensemble members. Concretely, let $\{(\alpha_m, \omega_m)\}_{m=1}^{M}$ be the ensemble members and their optimised weights, and $\{(\alpha_l, \omega_l)\}_{l=1}^{N-M}$ be the excluded architectures and their optimised weights. The weights of the ensemble $\boldsymbol{w} \in \mathbb{R}^M$ are given by

$$\boldsymbol{w}_m = \omega_m + \sum_{l=1}^{N-M} \frac{k(\alpha_m, \alpha_l)}{\sum_{m=1}^{M} k(\alpha_m, \alpha_l)} \omega_l. \tag{19}$$

---

**Algorithm 2** Posterior recombination.

---

$T \leftarrow \text{nystrom\_test\_functions}(K_{AA}, A)$          $\triangleright$ From Eq (11)

$\mu \leftarrow \left[ \frac{p(D|\alpha_n)p(\alpha_n)}{\hat{Z}} \right]_{n=1}^{N}$

$\boldsymbol{w}, A_M \leftarrow \text{recombination}(T, \mu)$

---

**Algorithm 3** Re-weighted stacking.

---

$\omega \leftarrow \text{argmin}_{\omega \in \Delta} \text{loss}(\sum_i \omega_i p(c \mid x, \alpha_n, D), D_{\text{val}})$

$I \leftarrow \text{select\_top}(M, \omega)$          $\triangleright$ Select top M.

**for** $m$ in I **do**

    $\boldsymbol{w}_m \leftarrow \text{reweight}(m, I, \omega, k(A, A))$          $\triangleright$ Eq (19).

**end for**

---

Our proposals can be combined to yield two possible algorithms. Both share the same candidate selection strategy that uses a WSABI-L surrogate model with the uncertainty sampling acquisition function to select the set of architectures to train (Algorithm 1). "BQ-R" then uses posterior recombination (Algorithm 2) to select a subset of architectures from the candidate set to include in the ensemble, and choose their corresponding weights. "BQ-S" instead uses re-weighted stacking (Algorithm 3 to select, and weight, the ensemble members from the candidate set. Note that "BQ-R" performs approximate hierarchical Bayesian inference using BQ, but "BQ-S" is a heuristic inspired by BQ. Figure 1 is a schematic representation of these algorithms.

## 4 Experiments

We begin by performing comparisons on the NATS-Bench benchmark (Dong et al., 2021). Specifically, we use the provided topology search space, which consists of cells with 4 nodes, 6 connections, and 5 possible operations (including "zeroise" which is equivalent to removing a connection) in a fixed macro-skeleton. The architecture weights are trained for 200 epochs on the CIFAR-100 and ImageNet16-120 (a smaller version of ImageNet with $16 \times 16$ pixel input images, and 120 classes) datasets. We will compare ensemble performance as measured by test accuracy, test likelihood, and expected calibration error on the test set for a range of ensemble sizes. The log-likelihood of the test set measures a model's performance both in terms of accuracy and uncertainty calibration, as placing a very low probability on the true class (i.e. being confidently wrong) is heavily penalised by this metric.

First, we verify that our chosen surrogate model (WSABI-L) performs well. Table 1 shows model performance, measured by root mean square error (RMSE) and negative log predictive density (NLPD) on a test set. The test set is selected by ranking all the architectures in the search space by validation loss, and selecting every 25th architecture. This ensures that the test set contains architectures across the full range of performance. We build on the results of Ru et al. (2021), who showed that a GP with a WL kernel is able to model the architecture likelihood surface well. Our results show that WSABI-L (with a WL kernel) is a consistently better model than an ordinary GP (with a WL kernel).

Next, we examine the effect of the candidate selection algorithm, shown in Table 2. In all cases, we use our variant of weighted stacking, described in Section 3.2, to select and weight the ensemble members. We compare Expected Improvement (EI) with a GP surrogate with a WL kernel, Uncertainty Sampling with a

| Model | CIFAR-100 | | ImageNet16-120 | |
|---|---|---|---|---|
| | RMSE | NLPD | RMSE | NLPD |
| GP | $6.165 \pm 0.116$ | $0.124 \pm 0.013$ | $9.610 \pm 0.626$ | $0.121 \pm 0.012$ |
| WSABI-L | $\mathbf{5.797 \pm 0.043}$ | $\mathbf{-2.741 \pm 0.095}$ | $\mathbf{4.078 \pm 0.058}$ | $\mathbf{-3.437 \pm 0.040}$ |

Table 1: The (normalised) RMSE and NLPD of a WSABI-L surrogate and a GP surrogate on the test sets.

| Algorithm | CIFAR-100 | | | ImageNet16-120 | | |
|---|---|---|---|---|---|---|
| | Accuracy | ECE | LL | Accuracy | ECE | LL |
| $M = 3$ | | | | | | |
| RE | $\mathbf{77.1 \pm 0.2}$ | $\mathbf{0.018 \pm 0.001}$ | $\mathbf{-4385 \pm 24.89}$ | $51.9 \pm 0.2$ | $\mathbf{0.029 \pm 0.002}$ | $-5595 \pm 12.15$ |
| EI | $76.1 \pm 0.2$ | $0.024 \pm 0.001$ | $-4472 \pm 29.74$ | $51.4 \pm 0.2$ | $0.034 \pm 0.002$ | $-5632 \pm 11.91$ |
| US | $76.6 \pm 0.2$ | $0.021 \pm 0.001$ | $\mathbf{-4417 \pm 35.85}$ | $\mathbf{52.2 \pm 0.1}$ | $\mathbf{0.029 \pm 0.001}$ | $\mathbf{-5543 \pm 10.87}$ |
| $M = 5$ | | | | | | |
| RE | $\mathbf{78.5 \pm 0.2}$ | $\mathbf{0.033 \pm 0.001}$ | $\mathbf{-4013 \pm 19.08}$ | $53.3 \pm 0.2$ | $\mathbf{0.043 \pm 0.002}$ | $-5417 \pm 12.90$ |
| EI | $77.4 \pm 0.2$ | $0.039 \pm 0.001$ | $-4126 \pm 22.25$ | $52.6 \pm 0.3$ | $0.053 \pm 0.003$ | $-5479 \pm 15.55$ |
| US | $77.8 \pm 0.2$ | $0.040 \pm 0.002$ | $-4077 \pm 33.60$ | $\mathbf{53.6 \pm 0.1}$ | $0.050 \pm 0.002$ | $\mathbf{-5380 \pm 12.31}$ |
| $M = 10$ | | | | | | |
| RE | $\mathbf{79.4 \pm 0.1}$ | $\mathbf{0.053 \pm 0.002}$ | $\mathbf{-3759 \pm 16.38}$ | $54.5 \pm 0.2$ | $\mathbf{0.065 \pm 0.002}$ | $-5280 \pm 16.85$ |
| EI | $78.2 \pm 0.2$ | $\mathbf{0.055 \pm 0.002}$ | $-3889 \pm 23.79$ | $53.4 \pm 0.2$ | $0.071 \pm 0.002$ | $-5368 \pm 19.47$ |
| US | $78.6 \pm 0.2$ | $0.059 \pm 0.001$ | $-3843 \pm 22.71$ | $\mathbf{54.7 \pm 0.1}$ | $0.072 \pm 0.001$ | $\mathbf{-5262 \pm 9.964}$ |

Table 2: Test accuracy, expected calibration error, and log-likelihood on CIFAR-100 and ImageNet16-120 for our candidate set selection method (US) and baselines. The numbers shown are means and standard error of the mean over 10 repeats. Each candidate set selection method is initialised with 10 random architectures, and used to build a set of 150 architectures. The ensemble is chosen and weighted using our variant of weighted stacking. We see that the RE candidate set performs best for CIFAR-100 and in terms of ECE for ImageNet16-120. The US candidate set performs best in terms of accuracy and LL for ImageNet16-120.

WSABI-L surrogate using a WL kernel (US), and Regularised Evolution (RE). We find that the US candidate set performs best for ImageNet16-120 in terms of accuracy and LL, but that the RE candidate set performs best for ECE on ImageNet16-120, and across all metrics for CIFAR-100.

We then move on to comparing the effect of the ensemble selection algorithm, shown in Table 3. In all cases, we use uncertainty sampling with a WSABI-L surrogate to build the candidate set. We initialise with 10 architectures randomly selected from a uniform prior over the search space, and use the acquisition function to build a set of 150 architectures. We compare beam search (BS), weighted stacking (WS), recombination of the approximate posterior (PR), and re-weighted stacking (RS). We find that the stacking variants consistently perform best (with RS slightly improving upon WS) in terms of accuracy and LL, and PR in terms of ECE for larger datasets.

We then proceed to compare the two variants of our algorithm – BQ-R and BQ-S – with several baselines.

**Random** The ensemble is an evenly weighted combination of $M$ architectures randomly sampled from the prior $p(\alpha)$ over the search space.

**Hyper-DE** The candidate set is selected by randomly sampling from the prior $p(\alpha)$. The ensemble is then chosen using beam search, with replacement.

**NES-RE** The candidate set is selected using regularised evolution, and the ensemble members are chosen using beam search. The ensemble members are equally weighted.

**NES-BS** The posterior over architectures $p(\alpha \mid D)$ is approximated using a variational distribution. The ensemble is constructed by sampling $M$ architectures from the variational distribution using Stein-Variational Gradient Descent. As no implementation is publicly available, we provide our own.

| | CIFAR-100 | | | ImageNet16-120 | | |
|---|---|---|---|---|---|---|
| Algorithm | Accuracy | ECE | LL | Accuracy | ECE | LL |
| *M* = 3 | | | | | | |
| BS | 75.2 ± 0.2 | 0.030 ± 0.002 | -4500 ± 41.04 | **52.2 ± 0.1** | 0.036 ± 0.002 | -5572 ± 13.17 |
| WS | **76.4 ± 0.2** | **0.021 ± 0.001** | **-4426 ± 35.87** | 52.1 ± 0.1 | **0.029 ± 0.001** | **-5545 ± 10.57** |
| PR | 71.9 ± 0.8 | 0.075 ± 0.025 | -5259 ± 300.9 | 46.7 ± 2.3 | 0.052 ± 0.021 | -6347 ± 480.5 |
| RS | **76.6 ± 0.2** | **0.021 ± 0.001** | **-4417 ± 35.85** | **52.2 ± 0.1** | **0.029 ± 0.001** | **-5543 ± 10.87** |
| *M* = 5 | | | | | | |
| BS | 76.4 ± 0.2 | 0.048 ± 0.002 | -4233 ± 36.48 | **53.4 ± 0.1** | 0.058 ± 0.002 | -5410 ± 10.70 |
| WS | **77.7 ± 0.2** | **0.036 ± 0.002** | **-4088 ± 34.13** | 53.6 ± 0.1 | 0.049 ± 0.001 | **-5382 ± 12.55** |
| PR | 73.3 ± 0.9 | **0.040 ± 0.004** | -4768 ± 174.3 | 50.7 ± 0.3 | **0.028 ± 0.004** | -5647 ± 50.58 |
| RS | **77.8 ± 0.2** | **0.040 ± 0.002** | **-4077 ± 33.60** | 53.6 ± 0.1 | 0.050 ± 0.002 | **-5380 ± 12.31** |
| *M* = 10 | | | | | | |
| BS | 76.9 ± 0.3 | 0.063 ± 0.001 | -4079 ± 50.29 | 54.1 ± 0.1 | 0.076 ± 0.001 | -5307 ± 9.795 |
| WS | **78.5 ± 0.2** | 0.055 ± 0.002 | **-3848 ± 23.96** | 54.6 ± 0.1 | 0.070 ± 0.001 | **-5264 ± 10.11** |
| PR | 75.5 ± 0.9 | **0.037 ± 0.002** | -4309 ± 172.6 | 52.3 ± 0.3 | **0.018 ± 0.001** | -5412 ± 22.96 |
| RS | **78.6 ± 0.2** | 0.059 ± 0.001 | **-3843 ± 22.71** | **54.7 ± 0.1** | 0.072 ± 0.001 | **-5262 ± 9.964** |

Table 3: Test accuracy, expected calibration error, and log-likelihood on CIFAR-100 and ImageNet16-120 for Beam Search (BS), Weighted Stacking (WS), Posterior Recombination (PR), and Re-weighted Stacking (RS). The numbers shown are means and standard error of the mean over 10 repeats. The candidate set selection method is our method – Uncertainty Sampling with a WSABI-L surrogate – initialised with 10 random architectures, and used to build a set of 150 architectures. We see that the stacking variants consistently perform best for accuracy and LL, with RS slightly improving upon WS. For ECE, RS and WS perform well for small ensembles, but PR works best for larger ensembles.

However, our implementation learns the variational parameters by approximating the expected log-likelihood term of the ELBO using a subset of the search space, rather than by backpropagating through a "supernet" as described by Shu et al. (2022). The subset we use is the 150 architectures in the search space with the highest likelihoods on the validation set. (Of course, this is only possible when working with a NAS benchmark.) We argue that our approximation is suitable as most posterior mass will be concentrated on these architectures, so a good variational distribution will concentrate mass on them as well. Additionally, our approximation is much faster as it does not require training a supernet.

Table 4 presents the results on CIFAR-100 and ImageNet16-120 for a range of ensemble sizes. Whilst NES-RE matches or does slightly better than our proposals in terms of accuracy and LL on CIFAR-100, we find that both BQ-S and BQ-R often perform better in terms of expected calibration error. BQ-S achieves the best performance on ImageNet16-120 in terms of LL across all ensemble sizes, is joint best with NES-RE in terms of accuracy, and often outperforms NES-RE in terms of ECE.

Next, we perform a study on a larger search space defined by a "slimmable network" (Yu et al., 2019), consisting of 614,625 architectures. Sub-networks or "slices" of this supernet constitute architectures within this search space. The architectures are structured as a chain of 7 blocks, each of which can have up to 4 layers. These sub-networks can be represented in a 28-dimensional ordinal space (with 4 options along each dimension). We compare the best-performing variant of our method, BQ-S, and the best-performing baseline, NES-RE, from the smaller NATS-Bench search space. We use an RBF kernel with WSABI-L for Uncertainty Sampling with our method BQ-S, and compare it to NES-RE. The results are shown in Table 5. We see that BQ-S consistently outperforms NES-RE in terms of the log-likelihood of the test set and, for CIFAR-100, in terms of expected calibration error as well.

Finally, we perform experiments to examine the robustness to dataset shift. Previous work has provided evidence that ensembling of Neural Networks provides robustness to shifts in the underlying data distribution (Zaidi et al., 2022; Shu et al., 2022). However, these investigations have assumed the availability of a validation set from the shifted distribution, which we argue is unrealistic in practice. Instead, we examine the setting where only the test set is shifted, and the validation set is representative of the training set. We

| Algorithm | CIFAR-100 | | | ImageNet16-120 | | |
|---|---|---|---|---|---|---|
| | Accuracy | ECE | LL | Accuracy | ECE | LL |
| Best Single | 69.1 | 0.088 | -5871 | 45.9 | 0.062 | -6386 |
| $M = 3$ | | | | | | |
| Random | 69.2 ± 1.5 | 0.075 ± 0.007 | -5778 ± 291.3 | 39.7 ± 2.2 | 0.097 ± 0.007 | -7459 ± 309.6 |
| Hyper-DE | 76.2 ± 0.2 | 0.030 ± 0.002 | -4390 ± 20.95 | 51.4 ± 0.2 | 0.040 ± 0.002 | -5659 ± 24.21 |
| NES-RE | **76.6 ± 0.2** | 0.026 ± 0.002 | **-4340 ± 19.58** | **52.0 ± 0.2** | 0.033 ± 0.002 | -5582 ± 8.858 |
| NES-BS | 66.2 ± 1.5 | 0.073 ± 0.009 | -6477 ± 203.0 | 45.7 ± 0.3 | 0.058 ± 0.003 | -6403 ± 28.04 |
| BQ-R | 71.9 ± 0.8 | 0.075 ± 0.025 | -5259 ± 300.9 | 46.7 ± 2.3 | 0.052 ± 0.021 | -6347 ± 480.5 |
| BQ-S | **76.6 ± 0.2** | **0.021 ± 0.001** | -4417 ± 35.85 | **52.2 ± 0.1** | **0.029 ± 0.001** | **-5543 ± 10.87** |
| $M = 5$ | | | | | | |
| Random | 72.2 ± 0.9 | 0.111 ± 0.009 | -5304 ± 180.9 | 42.7 ± 1.5 | 0.129 ± 0.008 | -7135 ± 216.1 |
| Hyper-DE | 77.6 ± 0.1 | 0.048 ± 0.002 | -4099 ± 12.67 | 52.4 ± 0.2 | 0.061 ± 0.001 | -5535 ± 22.38 |
| NES-RE | **78.2 ± 0.1** | **0.042 ± 0.002** | **-4002 ± 17.11** | **53.4 ± 0.2** | 0.051 ± 0.001 | -5404 ± 12.59 |
| NES-BS | 65.9 ± 1.5 | 0.073 ± 0.009 | -6481 ± 208.7 | 45.7 ± 0.3 | 0.058 ± 0.003 | -6403 ± 28.04 |
| BQ-R | 73.3 ± 0.9 | **0.040 ± 0.004** | -4768 ± 174.3 | 50.7 ± 0.3 | **0.028 ± 0.004** | -5647 ± 50.58 |
| BQ-S | 77.8 ± 0.2 | **0.040 ± 0.002** | -4077 ± 33.60 | **53.6 ± 0.1** | 0.050 ± 0.002 | **-5380 ± 12.31** |
| $M = 10$ | | | | | | |
| Random | 74.7 ± 0.3 | 0.150 ± 0.010 | -5018 ± 82.21 | 45.1 ± 0.4 | 0.159 ± 0.008 | -6916 ± 73.21 |
| Hyper-DE | 78.6 ± 0.1 | 0.066 ± 0.001 | -3862 ± 9.328 | 53.1 ± 0.2 | 0.075 ± 0.002 | -5466 ± 18.60 |
| NES-RE | **79.4 ± 0.1** | 0.060 ± 0.001 | **-3763 ± 15.16** | **54.5 ± 0.2** | 0.069 ± 0.001 | **-5269 ± 17.83** |
| NES-BS | 69.1 ± 0.4 | 0.085 ± 0.005 | -6119 ± 36.31 | 45.6 ± 0.3 | 0.068 ± 0.004 | -6442 ± 24.47 |
| BQ-R | 75.5 ± 0.9 | **0.037 ± 0.002** | -4309 ± 172.6 | 52.3 ± 0.3 | **0.018 ± 0.001** | -5412 ± 22.96 |
| BQ-S | 78.6 ± 0.2 | 0.059 ± 0.001 | -3843 ± 22.71 | **54.7 ± 0.1** | 0.072 ± 0.001 | **-5262 ± 9.964** |

Table 4: Test accuracy, expected calibration error (ECE), and log-likelihood (LL) on CIFAR-100 and ImageNet16-120 for our proposals (BQ-R and BQ-S) and baselines. For reference, we also include the performance of the best architecture (measured by validation loss) on the test set (labelled Best Single). The numbers shown are means and standard error of the mean over 10 repeats. Where applicable, the candidate set selection method is initialised with 10 random architectures and used to build a set of 150 architectures. For ImageNet16-120 we see that BQ-S performs best across ensemble sizes in terms of LL, and joint best with NES-RE in terms of accuracy. For CIFAR-100 we find that NES-RE performs best in terms of accuracy and LL. Particularly for larger ensembles, BQ-R performs best in terms of ECE.

| Algorithm | CIFAR-10 | | | CIFAR-100 | | |
|---|---|---|---|---|---|---|
| | Accuracy | ECE | LL | Accuracy | ECE | LL |
| $M = 3$ | | | | | | |
| NES-RE | 93.8 ± 0.0 | 0.029 ± 0.001 | -1165 ± 5.602 | 74.2 ± 0.2 | 0.072 ± 0.004 | -5136 ± 61.49 |
| BQ-S | 93.7 ± 0.1 | 0.030 ± 0.000 | **-1152 ± 5.215** | 74.4 ± 0.1 | **0.063 ± 0.002** | **-5021 ± 22.71** |
| $M = 5$ | | | | | | |
| NES-RE | 93.8 ± 0.0 | **0.030 ± 0.001** | -1165 ± 5.503 | 74.3 ± 0.2 | 0.071 ± 0.004 | -5134 ± 60.72 |
| BQ-S | 93.7 ± 0.1 | 0.032 ± 0.000 | **-1113 ± 4.123** | 74.5 ± 0.1 | **0.055 ± 0.002** | **-4897 ± 25.66** |
| $M = 10$ | | | | | | |
| NES-RE | 93.8 ± 0.0 | **0.030 ± 0.001** | -1159 ± 5.959 | 74.3 ± 0.2 | 0.069 ± 0.004 | -5083 ± 58.42 |
| BQ-S | 93.8 ± 0.0 | 0.031 ± 0.000 | **-1098 ± 3.752** | **74.7 ± 0.1** | **0.045 ± 0.001** | **-4766 ± 15.89** |

Table 5: Test accuracy, expected calibration error (ECE), and log-likelihood (LL) on CIFAR-10 and CIFAR-100 for BQ-S (our proposal) and NES-RE (the strongest baseline) for the "Slimmable Network" search space. We see that BQ-S consistently outperforms NES-RE in terms of ECE and LL, whilst maintaining the same accuracy.

use the benchmark established by Hendrycks & Dietterich (2019) to generate shifted datasets by applying one of 30 corruption types to each image for CIFAR-10 and CIFAR-100. Each corruption type has a severity level on a $1 - 5$ scale. Table 6 shows a comparison between NES-RE and BQ-S in this setting (on the

|  | CIFAR-10 | | | CIFAR-100 | | |
|---|---|---|---|---|---|---|
| Algorithm | Accuracy | ECE | LL | Accuracy | ECE | LL |
| $M = 3$ |  |  |  |  |  |  |
| NES-RE | 86.20 ± 0.04 | 0.046 ± 0.001 | -59259.6 ± 595.907 | 62.36 ± 0.08 | 0.151 ± 0.004 | -169235 ± 1632.43166 |
| BQ-S | 86.26 ± 0.08 | **0.036 ± 0.001** | **-54283.4 ± 642.383** | 62.52 ± 0.10 | **0.093 ± 0.002** | **-149022 ± 364.57480** |
| $M = 5$ |  |  |  |  |  |  |
| NES-RE | 86.25 ± 0.04 | 0.046 ± 0.001 | -59178.3 ± 851.719 | 62.36 ± 0.09 | 0.155 ± 0.003 | -169999 ± 1553.96433 |
| BQ-S | 86.16 ± 0.06 | **0.032 ± 0.001** | **-52173.6 ± 202.350** | **62.58 ± 0.09** | **0.103 ± 0.004** | **-152466 ± 1249.96873** |
| $M = 10$ |  |  |  |  |  |  |
| NES-RE | 86.26 ± 0.04 | 0.043 ± 0.001 | -57010.4 ± 722.311 | 62.46 ± 0.07 | 0.145 ± 0.002 | -164816 ± 975.78975 |
| BQ-S | 86.22 ± 0.05 | **0.029 ± 0.001** | **-50504.6 ± 443.984** | 62.54 ± 0.08 | **0.093 ± 0.002** | **-149022 ± 364.57480** |

Severity Level 1

|  | CIFAR-10 | | | CIFAR-100 | | |
|---|---|---|---|---|---|---|
| Algorithm | Accuracy | ECE | LL | Accuracy | ECE | LL |
| $M = 3$ |  |  |  |  |  |  |
| NES-RE | 73.16 ± 0.08 | 0.147 ± 0.002 | -133205 ± 1537.15 | 49.18 ± 0.07 | 0.235 ± 0.005 | -270710 ± 2628.75462 |
| BQ-S | 73.31 ± 0.12 | **0.131 ± 0.002** | **-123113 ± 1498.55** | **49.45 ± 0.12** | **0.193 ± 0.007** | **-250337 ± 3304.95** |
| $M = 5$ |  |  |  |  |  |  |
| NES-RE | 73.18 ± 0.09 | 0.148 ± 0.002 | -133239 ± 1904.50 | 49.20 ± 0.09 | 0.239 ± 0.004 | -272004 ± 2482.09961 |
| BQ-S | 73.23 ± 0.07 | **0.125 ± 0.001** | **-118756 ± 614.899** | **49.57 ± 0.09** | **0.175 ± 0.005** | **-241407 ± 2438.78** |
| $M = 10$ |  |  |  |  |  |  |
| NES-RE | 73.23 ± 0.08 | 0.143 ± 0.002 | -128663 ± 1664.07 | 49.29 ± 0.07 | 0.227 ± 0.003 | -263639 ± 1596.01214 |
| BQ-S | 73.39 ± 0.11 | **0.120 ± 0.002** | **-114613 ± 1247.44** | **49.57 ± 0.06** | **0.163 ± 0.003** | **-235152 ± 881.120** |

Severity Level 3

|  | CIFAR-10 | | | CIFAR-100 | | |
|---|---|---|---|---|---|---|
| Algorithm | Accuracy | ECE | LL | Accuracy | ECE | LL |
| $M = 3$ |  |  |  |  |  |  |
| NES-RE | 55.49 ± 0.08 | 0.285 ± 0.002 | -239927 ± 2187.24 | 34.04 ± 0.06 | 0.339 ± 0.005 | -415182 ± 3710.25 |
| BQ-S | 55.67 ± 0.14 | **0.265 ± 0.003** | **-226433 ± 2523.79** | **34.24 ± 0.11** | **0.291 ± 0.008** | **-385063 ± 5355.88** |
| $M = 5$ |  |  |  |  |  |  |
| NES-RE | 55.53 ± 0.08 | 0.286 ± 0.003 | -240154 ± 2835.89 | 34.04 ± 0.07 | 0.344 ± 0.005 | -417110 ± 3476.95 |
| BQ-S | 55.51 ± 0.05 | **0.260 ± 0.002** | **-220196 ± 1198.20** | **34.35 ± 0.10** | **0.270 ± 0.006** | **-371575 ± 4083.67** |
| $M = 10$ |  |  |  |  |  |  |
| NES-RE | 55.54 ± 0.08 | 0.279 ± 0.002 | -233441 ± 2474.23 | 34.11 ± 0.07 | 0.331 ± 0.003 | -405068 ± 2327.88 |
| BQ-S | 55.61 ± 0.11 | **0.254 ± 0.003** | **-214126 ± 2030.20** | **34.35 ± 0.07** | **0.257 ± 0.003** | **-361876 ± 1666.07** |

Severity Level 5

Table 6: Test accuracy, expected calibration error (ECE), and log-likelihood (LL) on CIFAR-10 and CIFAR-100 for NES-RE (the strongest baseline), and BQ-S (our strongest proposal) using the "Slimmable Network" search space for a range of corruption severities. We see that BQ-S is more robust than NES-RE to dataset shift, especially in terms of LL and ECE.

slimmable network search space). We see that, whilst our proposal performs similarly in terms of accuracy, it produces ensembles that perform significantly better in terms of expected calibration error and test set log-likelihood. This trend holds across corruption severity levels.

# 5 Discussion and Future Work

We proposed a method for building ensembles of Neural Networks using the tools provided by Bayesian Quadrature. Specifically, by viewing ensembling as approximately performing marginalisation over architectures, we used the warped Bayesian Quadrature framework to select a candidate set of architectures to train. We then suggest two methods of constructing the ensemble based upon this candidate set: one based upon recombination of the approximate posterior over architectures (BQ-R), and one based upon optimisation of the ensemble weights (BQ-S) using a validation set. BQ-R approximately performs hierarchical Bayesian inference using BQ, whereas BQ-S is a heuristic inspired by BQ. The discrepancy in performance is likely due to the fact that BQ-R does not make use of the validation set, as it takes the Bayesian perspective

and performs hierarchical inference over both architecture weights and architectures using the training set. (In principle, BQ-R can use the union of the training and validation sets to perform hierarchical inference. However, we did not run experiments in this setting as it would obviously allow BQ-R significantly more compute than the alternative methods.) For the same reason, BQ-R is more sensitive to any errors introduced by approximating the architecture likelihood using MLE (or any other approximation). BQ-S (and all the baselines), however, make use of a separate validation set to select the ensemble weights. We additionally show that BQ-S outperforms state-of-the-art baselines when the search space is large, and on the largest datasets for smaller search spaces. This is likely because it is more exploratory than alternative methods, and so less likely to become stuck near local minima of the architecture likelihood. Lastly, we demonstrated that BQ-S is more robust to dataset shift than state-of-the-art baselines.

A limitation of our proposals is that they do not outperform existing methods in some cases, notably on the CIFAR-100 dataset for the NATS-Bench search space. Additionally, it will be challenging to scale our proposals to larger evaluation budgets (greater than 1000) as the computational burden of inverting the GP's covariance matrix will become too large. An interesting direction for future work is to examine the effect of marginalising over architecture weights as well as over architectures.

We introduce a general-purpose method, so its societal impacts will depend on the specific tasks to which it is applied. We find it difficult to anticipate what those tasks will be, and even more difficult to speculate meaningfully about any societal impacts will be.

### Acknowledgments

The authors would like to thank anonymous reviewers for detailed and constructive feedback. S.H. and M.O are grateful for funding from the EPSRC and AIMS at the University of Oxford. S.H. is additionally supported by a scholarship from Kellogg College. X.W. and B.R. are supported by the Clarendon Scholarship at the University of Oxford. M.J. is supported by the Carlsberg Foundation.

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

## A  Computational Complexity

The computational complexity of candidate set selection strategies that require a GP surrogate (such as our proposals) is dominated by the cost of inverting the kernel matrix, which is, at each iteration, cubic in the size of the candidate set $\mathcal{O}(|A|^3)$. The complexity of re-weighted stacking is $\mathcal{O}\big(|A|\log(|A|) + |A_M|(|A| - |A_M|)\big)$. The first term is due to the sorting of candidate set, and the second to the computation the relevant covariances. The complexity of posterior recombination is dominated by the eigendecomposition of the kernel matrix, $\mathcal{O}(|A|^3)$.

Table 7 compares the run-time of our proposals against the most performant baseline on the NATS-Bench topology search space. Note that all architecture-related evaluations are cached (i.e. the trained weights are loaded from the NATS-Bench API, and the logits for the forward pass through the train/validation set are loaded from disk). The key reason that NES-RE is slower than our proposals is the iterative nature of Beam Search (required at each step of candidate set selection and then for ensemble selection). Recall that it greedily builds up the set of parent candidates (resp. final ensemble) by iterating through the whole pool (resp. population). Each iteration requires loading a set of logits from disk which, whilst cheaper than loading and performing a forward pass through the architecture, still incurs significant computational cost in aggregate.

Note that, for most search spaces, the computational cost of Neural Ensemble Search is dominated by the cost of evaluating the likelihoods of the architectures for the candidate set. For some spaces, such as those defined by a supernet, this cost is lower as the supernet is only trained once. However, this initial training is still computationally intensive. Therefore, in either case, the computation related to training and evaluating architectures is likely to be significantly larger than the computation required for the Neural Ensemble Search method.

## B  Further Analysis of Results

Our results over the NATS-Bench topology search space show that NES-RE performs best for the CIFAR-100 dataset, and BQ-S performs best for ImageNet16-120 (see Table 4). The ablation study in Tables 2 and 3 suggest that this is due to the candidate selection strategy. Recall that NES-RE uses regularised

| Algorithm | CIFAR-100 | ImageNet16-120 |
|---|---|---|
| NES-RE | $24236.3 \pm 109.551$ | $12020.6 \pm 98.1485$ |
| BQ-R | $308.872 \pm 7.05121$ | $258.444 \pm 3.76138$ |
| BQ-S | $348.650 \pm 11.7402$ | $280.689 \pm 9.76650$ |

Table 7: The runtime of our proposals, BQ-R and BQ-S, against the most competitive (in terms of performance) baseline on CIFAR-100 and ImageNet16-120 over the NATS-Bench topology search space. The total evaluation budget is 150 architectures, and we select ensembles of size 5. We report the means and standard error of the means over 3 runs. All architecture-related evaluations are cached (i.e. the trained weights are loaded from the NATS-Bench API, and the logits for the forward pass through the train/validation set are loaded from disk).

evolution for candidate set selection and beam search for ensemble selection, whereas BQ-S uses uncertainty sampling with a WSABI-L surrogate and re-weighted stacking. Table 2 suggests that (for a fixed ensemble selection strategy) regularised evolution is the best method for CIFAR-100 but that uncertainty sampling with a WSABI-L surrogate is the best method for ImageNet16-120. Table 3 shows that re-weighted stacking consistently outperforms beam search given a fixed candidate set. From these results, we can infer that regularised evolution is a better candidate selection strategy for CIFAR-100 and uncertainty sampling is better for ImageNet16-120. This must be due to the nature of the architecture likelihood surfaces for each dataset. As these surfaces are defined over an input space of architectures, they are difficult to visualise. One possible method is shown in Figure 2. We consider the 500 architectures with the highest likelihood. We then visualise the covariance matrix between them using the WL kernel of a trained GP. The architectures are sorted using the GP's estimate of the likelihood (to smooth out noise). We observe larger clusters of architectures along the diagonal of the covariance matrix for CIFAR-100. This means that they covary strongly and have similar likelihoods. Further, note that these clusters covary strongly with each other. This provides some evidence that the peaks of the CIFAR-100 likelihood surface are broader and closer together (based on the metric implied by the WL kernel) than those for the ImageNet16-120 likelihood surface. On this basis, we suggest that uncertainty sampling with a WSABI-L surrogate is better suited to architecture likelihood surfaces with dispersed, narrow peaks and regularised evolution is better suited to architecture likelihood surfaces with wider peaks.

## C  Additional Uncertainty Metrics

To verify that the quality of the uncertainty is well measured by the expected calibration error we check that it correlates well with the calibration AUC (Kivlichan et al., 2021; Rožanec et al., 2023). These results are shown in Table 8 (for the same experimental setup as in Table 4). Note that well-calibrated uncertainty is indicated by high calibration AUC but low expected calibration error. The correlation between the (means of the) two metrics is -0.71 for CIFAR-100 and -0.20 for ImageNet16-120. They are, therefore, generally in agreement about the quality of a model's calibration estimates.

## D  Architecture Likelihood Approximation

We examine the impact of the approximations suggested in Equations 15 and 16 by comparing their performance to an alternative – Stochastic Weight Averaging (Gaussian) (Maddox et al., 2019). This instead approximates the posterior over architecture weights using a diagonal Gaussian, whose moments are the empirical mean and variance of several SGD iterates, obtained by continuing training. The results are shown on CIFAR-100 for the NATS-Bench search space in Table 9. We see significant improvements across all metrics when using the SWAG approximation, suggesting that there is much to be gained by marginalising over both architecture weights and architectures. This would be a promising direction for future work.

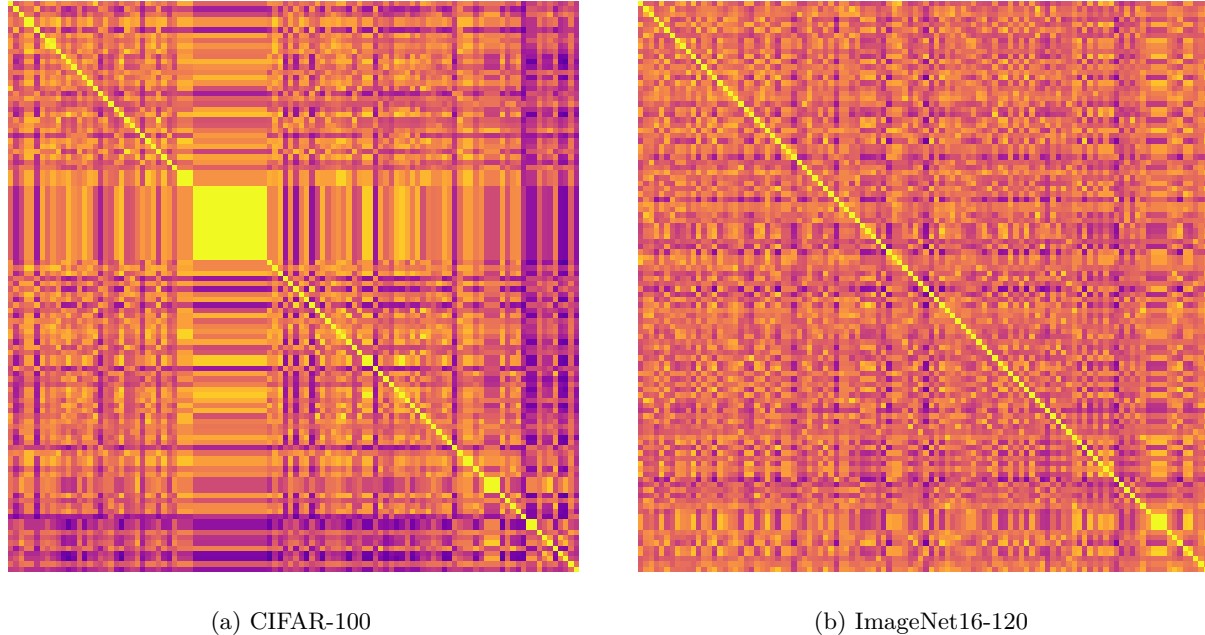

(a) CIFAR-100                 (b) ImageNet16-120

Figure 2: Visualisation of the (WL) covariance matrix for the 500 architectures with the highest likelihoods in the search space for each dataset, sorted by (a smoothed estimate, using a GP, of the) likelihood. The colour scale varies from 1 (yellow) to 0 (blue). We observe larger blocks of architectures within the top 500 that covary strongly for CIFAR-100 than for ImageNet16-120, which implies that the modes of the architecture likelihood surface are wider for CIFAR-100. This suggests that a more exploratory strategy will do better on ImageNet16-120, and a more exploitative strategy for CIFAR-100.

## E   Experimental Setup

The codebase uses PyTorch (Paszke et al., 2019) to handle deep learning and backpropagation. Except where otherwise stated, the experimental results report means and standard error of the mean over 10 repeats. Where applicable, each candidate selection method is initialised with 10 architectures randomly selected from the search space, and allowed to select an additional 140.

Our proposal over the cell-based search space uses the WL kernel, with its level hyperparameter chosen from $\{1, 2\}$ using the GP's marginal likelihood. For the macro-based search space, our proposal uses an ARD RBF kernel, whose hyperparameters are optimised using LBFGS. The lengthscales are constrained between the minimum and maximum distances between observations along the relevant dimensions. Architecture likelihoods are normalised so that the maximum observed is 1 before modelling with the GP surrogate. The noise hyperparameter is also selected to optimise the probability density assigned to the observed data under the GP prior. It is constrained in the range $[10^{-5}, 10^{-1}]$. The acquisition function is always optimised using an evolutionary strategy, using a pool size of 1024. Per iteration, we allow 128 mutations, of which 16 are modifications of the architecture with the highest acquisition value, and the remainder are selected uniformly at random from the pool.

| Algorithm | CIFAR-100 | | ImageNet16-120 | |
| --- | --- | --- | --- | --- |
| | ECE | C-AUC | ECE | C-AUC |
| $M = 3$ | | | | |
| NES-RE | $0.026 \pm 0.002$ | $0.873 \pm 0.002$ | $0.033 \pm 0.002$ | $0.823 \pm 0.002$ |
| NES-BS | $0.073 \pm 0.009$ | $0.848 \pm 0.003$ | $0.058 \pm 0.003$ | $0.809 \pm 0.003$ |
| BQ-R | $0.075 \pm 0.025$ | $0.866 \pm 0.002$ | $0.052 \pm 0.021$ | $0.815 \pm 0.003$ |
| BQ-S | $0.021 \pm 0.001$ | $0.874 \pm 0.001$ | $0.029 \pm 0.001$ | $0.824 \pm 0.002$ |
| $M = 5$ | | | | |
| NES-RE | $0.042 \pm 0.002$ | $0.875 \pm 0.001$ | $0.051 \pm 0.001$ | $0.826 \pm 0.002$ |
| NES-BS | $0.073 \pm 0.009$ | $0.848 \pm 0.003$ | $0.058 \pm 0.003$ | $0.809 \pm 0.003$ |
| BQ-R | $0.040 \pm 0.004$ | $0.871 \pm 0.001$ | $0.028 \pm 0.004$ | $0.820 \pm 0.002$ |
| BQ-S | $0.040 \pm 0.002$ | $0.877 \pm 0.001$ | $0.050 \pm 0.002$ | $0.825 \pm 0.002$ |
| $M = 10$ | | | | |
| NES-RE | $0.060 \pm 0.001$ | $0.877 \pm 0.001$ | $0.069 \pm 0.001$ | $0.825 \pm 0.001$ |
| NES-BS | $0.085 \pm 0.005$ | $0.848 \pm 0.001$ | $0.068 \pm 0.004$ | $0.808 \pm 0.002$ |
| BQ-R | $0.037 \pm 0.002$ | $0.872 \pm 0.001$ | $0.018 \pm 0.001$ | $0.819 \pm 0.002$ |
| BQ-S | $0.059 \pm 0.001$ | $0.879 \pm 0.001$ | $0.072 \pm 0.001$ | $0.826 \pm 0.002$ |

Table 8: Expected calibration error (ECE) and calibration AUC (C-AUC) on CIFAR-100 and ImageNet16-120 for our proposals (BQ-R and BQ-S) and baselines. Well-calibrated uncertainty is indicated by high calibration AUC but low expected calibration error. We see that the two metrics are generally in agreement about the quality of a model's uncertainty estimates.

| Algorithm | CIFAR-100 | | |
| --- | --- | --- | --- |
| | Accuracy | ECE | LL |
| $M = 3$ | | | |
| BQ-R (MLE) | $71.9 \pm 0.8$ | $0.075 \pm 0.025$ | $-5259 \pm 300.9$ |
| BQ-R (SWAG) | $75.1 \pm 0.6$ | $0.035 \pm 0.007$ | $-4516 \pm 140.2$ |
| BQ-S (MLE) | $76.6 \pm 0.2$ | $0.021 \pm 0.001$ | $-4417 \pm 35.85$ |
| BQ-S (SWAG) | $77.4 \pm 0.3$ | $0.022 \pm 0.000$ | $-4345 \pm 38.19$ |
| $M = 5$ | | | |
| BQ-R (MLE) | $73.3 \pm 0.9$ | $0.040 \pm 0.004$ | $-4768 \pm 174.3$ |
| BQ-R (SWAG) | $75.0 \pm 1.1$ | $0.039 \pm 0.012$ | $-4335 \pm 133.4$ |
| BQ-S (MLE) | $77.8 \pm 0.2$ | $0.040 \pm 0.002$ | $-4077 \pm 33.60$ |
| BQ-S (SWAG) | $78.8 \pm 0.4$ | $0.033 \pm 0.001$ | $-3944 \pm 25.47$ |
| $M = 10$ | | | |
| BQ-R (MLE) | $75.5 \pm 0.9$ | $0.037 \pm 0.002$ | $-4309 \pm 172.6$ |
| BQ-R (SWAG) | $77.9 \pm 0.2$ | $0.063 \pm 0.006$ | $-3964 \pm 8.734$ |
| BQ-S (MLE) | $78.6 \pm 0.2$ | $0.059 \pm 0.001$ | $-3843 \pm 22.71$ |
| BQ-S (SWAG) | $79.9 \pm 0.1$ | $0.053 \pm 0.001$ | $-3680 \pm 18.40$ |

Table 9: Test accuracy, expected calibration error (ECE), and log-likelihood (LL) on CIFAR-100 for BQ-R and BQ-S with MLE and SWAG approximations for the architecture likelihood. Note that, due to their computation expense, results for the SWAG approximations are means and standard error of the mean over 2 runs. For the MLE approximations, we report the mean and standard error of the mean over 10 runs.

