# OpenReview forum: "Bayesian Quadrature for Neural Ensemble Search"
_TMLR — Accepted by TMLR_

### Review · Reviewer_56g8 · 2023-04-19

**Summary Of Contributions:**

The authors consider the problem of Neural Ensemble Search (NES), that is, Neural Architecture Search over the space of neural network ensembles. They phrase this as a Bayesian Quadrature (BQ) problem, in which the selected ensemble members are used to approximate the marginal predictive over all architectures. They show that using acquisition functions from BQ can help with this problem and also propose a new recombination strategy for model weighting.

**Audience:**

Yes

**Broader Impact Concerns:**

No broader impact concers.

**Claims And Evidence:**

Yes

**Requested Changes:**

- Add a related work section
- Add some baselines (see above)
- Add runtimes to the tables.

**Strengths And Weaknesses:**

Strengths:
- The paper is clearly written.
- NES is an important problem.
- Using ideas from BQ is a creative approach and seems to be useful.

Weaknesses:
- Related work is missing.
- Similarly, some important baselines are missing.
- There is little discussion and empirical results on the computational cost / runtimes.

Major comments:
- In the Algorithm boxes, you use a lot of shorthand functions (acquisition_function, nystrom_test_function, reweight, etc). I think it would be more useful to write these out in maths, so one can see at one glance what is happening.
- While the experiments provide useful ablations between the different strategies, I feel like they are missing more common baselines. I would expect to see a standard deep ensemble (with the fixed largest architecture from the search space), a hyper-deep ensemble [1] and maybe some kind of diversity-regularized ensemble [e.g., 2-8].
- The paper is entirely missing a related work section. What have people done before to do NES? How does it compare to existing NAS approaches? What should at least be mentioned are the very related hyper-deep ensembles [1] and probably some examples of other diversity-regularized ensembles [e.g., 2-8].

Minor comments:
- In Tab. 2, why do US and RE perform differently on ImageNet and CIFAR? Do they encourage different kinds of architectures or amounts of diversity? It would be interesting to examine this more closely.
- How do the methods compare in terms of runtime? It would be useful for practitioners to know the tradeoffs, so reporting wallclock or GPU runtimes in all experiments would be helpful.

[1] https://arxiv.org/abs/2006.13570

[2] https://arxiv.org/abs/1806.03335

[3] https://arxiv.org/abs/2206.03633

[4] https://openreview.net/forum?id=BJlahxHYDS

[5] https://proceedings.neurips.cc/paper/2020/hash/0b1ec366924b26fc98fa7b71a9c249cf-Abstract.html

[6] https://arxiv.org/abs/1802.07881

[7] https://arxiv.org/abs/2106.10760

[8] https://proceedings.neurips.cc/paper/2021/hash/1c63926ebcabda26b5cdb31b5cc91efb-Abstract.html

---

> ### Author Response · Authors · 2023-05-11
>
> Thank you for your thoughtful feedback. We have uploaded a revised version of the manuscript where we have:
> - Added additional references and explanation of related work in Section 2.3.
> - Included a discussion of computational complexity in Appendix A.
>
> In addition, we are currently running the additional baseline, hyper-deep ensembles, and will include these results once available.
>
> Where we have made changes in the main paper we highlight this using red text.

---

### Review · Reviewer_M93y · 2023-04-27

**Summary Of Contributions:**

This paper uses Bayesian Quadrature to construct ensembles of neural networks. The key observation is that constructing an ensemble of neural networks can be seen as marginalizing over different neural network architectures and that this marginalization can be approximated using quadrature. This is analogous to the way in which neural architecture search can be seen as an optimization problem over the space of different architectures that can be approximated using Bayesian optimization. The paper views ensemble construction as a two-step process. Firstly, a set of candidate architectures is selected using an acquisition function. Secondly, a (weighted) ensemble is constructed from a subset of the candidate architectures. The first problem is solved with a novel use of Bayesian Quadrature. Two potential methods are proposed for solving the second problem, each of which is better in some settings.

**Audience:**

Yes

**Claims And Evidence:**

No

**Requested Changes:**

I use [major] and [minor] to indicate the changes that would be crucial to securing my recommendation or would simply strengthen the work, respectively. Although most clarity issues are listed individually as [minor], they represent a more major issue when taken together.

1. [major] Add a citation (or other evidence) for the claim that "... existing approaches to ensembling struggle when the architecture likelihood surface has dispersed, narrow peaks".
2. [major] Add a citation for the claim that Bayesian Quadrature is "... well suited to exploring likelihood surfaces with dispersed, narrow peaks".
3. [minor] Add a discussion of Wenzel et al. (2020), alongside Zaidi et al. (2022) and Shu et al. (2022) when discussing the idea that ensembles of different architectures outperform ensembles of the same architecture. Wenzel et al. (2020) show that ensembles with different hyperparameters (which could include architectural hyperparameters) outperform standard ensembles.
4. [minor] The claim that single architectures provide poor uncertainty estimates is not well supported. Firstly, Guo et al. (2017) focus on a narrow range of architectures (namely CNNs), and a follow-up study (Minderer et al., 2021) shows that this is not true for other architectures (e.g. ViTs). Secondly, several techniques can be applied to single models to improve calibration. I would change this statement and instead say that ensembles tend to be better calibrated than single models.
5. [major] Add additional uncertainty metrics. I recommend using Calibration AUC (Kivlichan et al., 2021, Tran et al., 2022), which does not suffer from the same pathologies as ECE. Oracle Collaborative AUC and OOD detection AUC could also be a good choices. I acknowledge that adding these additional metrics to all experiments might require a large amount of work. However, I think that adding just Calibration AUC to only Table 4 would be good enough as if there is a good correlation between ECE and Calibration AUC there, then we can assume that the ECE results are reliable elsewhere.
6. [minor] Section 2.5 must be better integrated into the rest of the text. Reading the paper for the first time, this section seemed very arbitrary, and I was unsure how this would connect to the rest of the work. Even after a second reading, I felt it would be helpful to make connections between this section and the rest of the paper more explicit.
7. [minor] Add a legend to figure 1. What do the different colors mean?
8. [minor] On page 6, 3rd paragraph, the sentence "Therefore, we propose using a Bayesian Quadrature acquisition function to build the candidate set ...", did not follow from the previous sentence. Adding additional explanation and intuition would be helpful.
9. [major] Add an ablation study to show that the approximations in equations (14) and (16) are reasonable. Even in a small-scale setting and using some approximate inference scheme (I would suggest linearised Laplace), it would be useful to compare the performance of the proposed method with and without marginalizing the weights in (14) and (16). Otherwise, it is impossible to know what kind of impact the approximation has and how close the proposed method is in practice.
10. [minor] I think it would be useful to be more explicit about the different levels of Bayesian inference that are taking place. There is a first level in which we do inference over weights. And a second level where we do inference over architectures. The evidence of the first level is the likelihood of the second level. Chapter 28 of "Information Theory, Inference, and Learning Algorithms" by David J.C. MacKay explains this well.
11. [minor] The statement "computation of this estimate requires Monte-Carlo sampling to approximate sums of (products of) the WL kernel over A" needs to be made more explicit by writing out the MC sampling equation.
12. [minor] A few preliminary results showing the impact of using the union of the train and validation sets for BQ-R would be an interesting and useful addition.
13. [minor] Add a short discussion of the limitation of the method.
14. [major] Add an appendix section which describes the experimental setup in detail. Details such as which deep learning framework, how many random seeds for each experiment, and any hyper-parameter sweeps and settings should be included.



**References**

* Florian Wenzel, Jasper Snoek, Dustin Tran, Rodolphe Jenatton: Hyperparameter Ensembles for Robustness and Uncertainty Quantification. NeurIPS 2020

* Matthias Minderer, Josip Djolonga, Rob Romijnders, Frances Hubis, Xiaohua Zhai, Neil Houlsby, Dustin Tran, Mario Lucic:
Revisiting the Calibration of Modern Neural Networks. NeurIPS 2021: 15682-15694

* Ian D. Kivlichan, Zi Lin, Jeremiah Z. Liu, Lucy Vasserman:
Measuring and Improving Model-Moderator Collaboration using Uncertainty Estimation. CoRR abs/2107.04212 (2021)

* Dustin Tran, Jeremiah Z. Liu, Michael W. Dusenberry, Du Phan, Mark Collier, Jie Ren, Kehang Han, Zi Wang, Zelda Mariet, Huiyi Hu, Neil Band, Tim G. J. Rudner, Karan Singhal, Zachary Nado, Joost van Amersfoort, Andreas Kirsch, Rodolphe Jenatton, Nithum Thain, Honglin Yuan, Kelly Buchanan, Kevin Murphy, D. Sculley, Yarin Gal, Zoubin Ghahramani, Jasper Snoek, Balaji Lakshminarayanan:
Plex: Towards Reliability using Pretrained Large Model Extensions. CoRR abs/2207.07411 (2022)

**Strengths And Weaknesses:**

## Strengths

+ **Novelty:** While novelty is not an acceptance criterion for TMLR, this paper does propose a novel and interesting approach to ensemble construction. I suspect that many people would find this work stimulating and that it could lead to further interesting work in the future.

+ **Solid Experiments:** The experimental evaluation seems solid, and most of the claims are well supported.

## Weaknesses

- **Clarity:** I found the paper hard to understand in several places. The main issue is that the authors do not guide the reader through a few key sections. As a result, many statements seem arbitrary and do not follow previous statements, and I struggled to gain insight into some parts of the method. Making connections more explicit and assuming less knowledge/understanding on the reader's part would help greatly. I make concrete suggestions in the next section.

- **Evidence:** There are a small number of key statements for which I do not believe there is sufficient evidence in the paper.
   1. Existing approaches to ensembling struggle when the architecture likelihood surface has dispersed and narrow peaks.
   2. Bayesian Quadrature is well suited to exploring such likelihood surfaces.
   3. The proposed method improves calibration. The paper uses two metrics–LL and ECE–as measures of calibration. However, LL is not an ideal choice because it combines both accuracy and uncertainty calibration, making it difficult to disentangle where the improvement is coming from. Furthermore, in almost all cases, when LL improves, so does the accuracy, which does not help. ECE, on the other hand, is not entangled with accuracy but has several other pathologies. For example, a classifier can get perfect ECE while predicting only the marginal probabilities for each class. Another issue is that it is not obvious how ECE should be applied in a multi-class setting.
  4. Bayesian Quadrature is being used to construct the ensemble. Or rather, the Bayesian Quadrature used to construct the ensemble is well approximated. My concern is the approximations used in equations (14) and (16). These seem like very strong approximations, making it unclear whether the proposed method is doing something close to proper quadrature or if we should view the method as a well-performing heuristic inspired by quadrature.

---

> ### Author Response · Authors · 2023-05-11
>
> Thank you for your detailed and constructive feedback. We have uploaded a revision of the manuscript which makes the changes you have requested. Specifically, the updated manuscript:
> - Includes a section justifying our claims about the suitability of various methods for different likelihood surfaces in Appendix B.
> - Modifies the claim that single architectures provide poor uncertainty estimates to state that this is only in some cases.
> - Includes a comparison of ECE and Calibration AUC in Appendix C.
> - Adds introductory detail to Section 2.5 to better situate it in the text.
> - Improves the introduction of the idea of using a BQ acquisition function for candidate set selection on page 6, paragraph 3.
> - Describes the limitations of our proposals in Section 5, paragraph 2.
> - Includes a section describing the details of the experimental setup in Appendix E.
>
> Where the changes refer to text in the main paper we have indicated this using red text.

---

> > ### Comment · Reviewer_M93y · 2023-05-18
> > **Looking good!**
> >
> > Thanks for all of the updates and clarifications. In light of the changes, I am happy that both of the TMLR acceptance criteria are now met, and I will recommend acceptance. I would still encourage the authors to make the text as clear as possible. In particular, I would try to elicit feedback from a range of readers with different background knowledge levels (e.g., readers without much GP or NAS knowledge), as I think that while everything is correct, there could be more intuition and guidance to help readers understand the proposed method.

---

### Review · Reviewer_bZZQ · 2023-05-04

**Summary Of Contributions:**

This work proposes using Bayesian quadrature (BQ) for neural ensemble search (NES), which is a natural counterpart of using Bayesian optimization for neural architecture search (NAS).  The authors propose two approaches for NES based on a Gaussian process model over the architectural DAGs: a more faithful implementation to BQ (BQ-R), and a stacking variant (BQ-S) that reweight the learned ensemble using additional validation samples.  Across several benchmarks constructed out of the CIFAR and ImageNet datasets, the BQ-R method outperforms recent NES baselines in terms of uncertainty-related metrics.


**Audience:**

Yes

**Claims And Evidence:**

Yes

**Requested Changes:**


- (Enhancement) More discussion about the performance of BQ-R.

- (Enhancement) In Table 3 and 5 the BQ-R method is only competitive in the log likelihood (LL) metric.  The authors argued that the ECE and LL metrics are "crucial for systems which make critical decisions", but this is only obviously true for ECE (or other metrics that directly reflect calibration error), so there should be more discussions around the use of the LL metric.


**Strengths And Weaknesses:**


**Strengths:**

+ The idea of using Bayesian quadrature for NES is quite sensible and may deserver further investigation.

+ A variant of the BQ approach demonstrates promising empirical performance.

**Weaknesses:**

- The BQ-R method, which is closer to a full implementation of Bayesian quadrature, performs worse than most baselines. Thus, the evidence for the efficacy of BQ is somewhat limited.

- In the implementation, two marginalization operations over the NN parameters are replaced with an MLE approximation.  This is understandable given the increased computational cost of the latter, but it deviates from the BQ method and may be related to the previous point, the worsened performance of BQ-R.  The authors argued that is because the baselines have access to validation samples, but I think there should also be discussion around the MLE approximations, since (in many related settings) a fully Bayesian approach would not be hindered by the lack of validation samples.

(Note that I'm not familiar with the recent empirical results on NAS and NES, and my evaluation of the experiments is purely based on the authors' report.)

---

> ### Author Response · Authors · 2023-05-11
>
> Thank you for your considered review. We have uploaded a revised version of the manuscript that addresses your concerns. In particular, we have:
> - Added an additional suggestion for the performance discrepancy between BQ-R and BQ-S in Section 5.
> - We have explicitly discussed the advantages of the LL metric in the first paragraph of Section 4.
>
> We have highlighted the changes in red text so that they are easy to see.

---

### Decision · Action_Editors · 2023-07-05

**Recommendation:** Accept as is

**Comment:**

The paper introduces a novel approach for Neural Ensemble Search using Bayesian Quadrature. A number of issues were raised, but the main ones concerning the method itself focused on the approximations involved (BQ-R vs BQ-S)  and the use of log-likelihood to make claims about modelling uncertainty. There was relatively little discussion needed as the author's subsequent draft resolved all of these issues along with many other minor ones. The reviewers unanimously agreed that the paper should be accepted to TMLR.

**Audience:**

Yes, reviewer M93y put it well when they said "I suspect that many people would find this work stimulating and that it could lead to further interesting work in the future."

**Claims And Evidence:**

Both Reviewers bZZQ and M93y had issues with the use of log-likelihood as a metric for evaluating uncertainty. Reviewer M93y also had issues with a few of the claims made in the paper about the likelihood surface and whether the approach is really doing BQ after approximations, or something else. Reviewer 56g8 suggested that more baselines should be added. These were all addressed in a subsequent draft of the paper, to the satisfaction of the reviewers.